# SPECTRUM-AWARE SPIKING DYNAMICS FOR GRAPH CONTRASTIVE LEARNING

## ABSTRACT

Graph Contrastive Learning (GCL) typically relies on Graph Neural Networks (GNNs) for full-precision representation learning, which results in high computational overhead and energy consumption. Recently, integrating Spiking Neural Networks (SNNs) with GNNs has emerged as a promising energy-efficient alternative. However, existing approaches often treat spiking neurons merely as binary encoders to produce 1-bit representations, ignoring the rich structural information inherent in graph data. Also, the usage of a fixed initial membrane potential (IMP), which is usually set to 0, restricts the diversity of spiking patterns and limits the expressive power of spiking neurons. To address these issues, we propose a novel **S**pectrum-enhanced **S**piking **G**raph **C**ontrastive **L**earning (S$^2$GCL) framework by integrating graph spectral information into spiking dynamics. Specifically, we first develop a novel *Spectrum-aware Membrane Potential (SaMP)* mechanism for SNNs by injecting eigenvalue-based biases into membrane potential learning to capture global graph structure and enhance SNN's expressive power. Then, we introduce an *Overlapped Channel Grouping (OCG)* strategy to construct sequence spikes for the graph and simultaneously reinforce correlations in spike trains based on overlapped feature channels. Finally, we adopt the dual-level contrastive objective to achieve both node-wise and channel-wise alignments. Extensive experiments on several benchmark datasets show the effectiveness of our proposed S$^2$GCL. The code of our method will be released upon acceptance.

## 1 INTRODUCTION

Graph Contrastive Learning (GCL) is a mainstream research direction in self-supervised graph representation learning. It aims to learn graph encoders by maximizing consistency between different augmented graphs (Ju et al., 2024). Recently, GCL methods have shown state-of-the-art performance on many downstream tasks and many variants have been developed from the perspective of graph augmentation (Zhu et al., 2020; 2021; Xu et al., 2025) and contrastive objective (Zhang et al., 2021; Zheng et al., 2022). Despite promising results, the growing scale of real-world graphs (Zhang et al., 2024; 2025) has posed new challenges to GCL beyond the context of label-free learning. It is known that current GCL methods mainly require large hidden dimensions to learn full-precision representations, leading to high computational and memory overhead (Li et al., 2024). This inefficiency critically bottlenecks resource-constrained applications, such as edge devices, where energy consumption and storage cost are critical concerns.

For energy-efficient alternatives, Spiking Neural Networks (SNNs) have gained attention due to their event-driven computation and inherent sparsity (Han et al., 2020). Recent studies have explored the integration of SNNs and GNNs and developed Spiking Graph Neural Networks (SGNNs) (Zhu et al., 2022; Li et al., 2023; Zhao et al., 2024; Sun et al., 2024; Li et al., 2024). However, existing SGNNs suffer from the following two main limitations. **First**, Spiking GNNs often regard SNNs as a binarized learning model, directly converting the continuous representation to 1-bit spikes. This design fails to exploit the intrinsic properties of graph data (e.g. spectral or positional information) which are crucial cues for GCL process (Figure 1(a)). **Second**, SNNs generally adopt a fixed initial membrane potential (IMP) for each firing step, which limits the diversity of SNNs firing patterns and thus has limited discriminative and expressive capacity (Shen et al., 2024). Recent studies demonstrate that adjusting IMP can enable more pattern mappings (Shen et al., 2024), as shown in Figure 1(b). Therefore, an intuitive idea naturally arises: *can we develop a novel spiking graph*

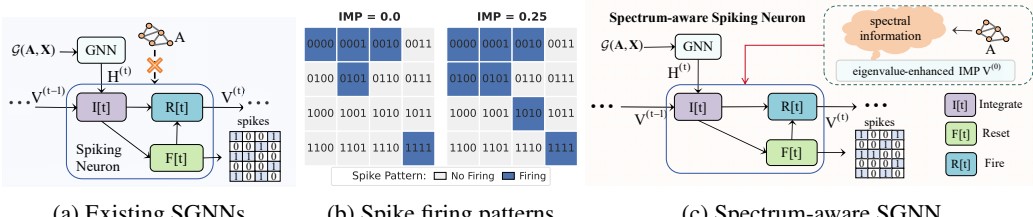

(a) Existing SGNNs     (b) Spike firing patterns     (c) Spectrum-aware SGNN

Figure 1: (a) Existing SGNNs simply use spike neurons for binary encoding. (b) All firing patterns (16 total) that IF model can generate under 4 time steps, where the blue boxes indicate the patterns can be fired, while the gray ones represent the patterns cannot be fired. New patterns emerge by adjusting IMP under constant intensity input. (c) Motivated by (a)-(b), we develop a novel Spectrum-aware SGNN by introducing graph spectral property to guide IMP dynamic learning.

*contrastive learning framework by introducing spectral information to induce membrane potential dynamics?*

In this paper, we propose to explicitly incorporate graph information into spiking neural dynamics and develop a novel **S**pectrum-inspired **S**piking **G**raph **C**ontrastive **L**earning ($S^2$GCL) framework. Specifically, we first design a novel SNN based on Spectrum-aware Membrane Potential (SaMP) by introducing graph spectral encoding for IMP dynamic learning. The core aspect of SaMP is that we achieve the membrane potential dynamic by embedding spectral-based inductive biases to capture global structural information of graph data and simultaneously enhance SNN's expressive capacity. Then, we design a new Overlapped Channel Grouping (OCG) strategy to construct spike trains in the channel space. The overlapped design can model the dependencies of channels and enhances feature interactions among different spike sequences. Finally, we develop a dual-level contrastive objective by combining node-wise and channel-wise contrastive alignment together. Overall, the main contributions of this paper are summarized as follows:

- We propose Spectrum-enhanced Spiking Graph Contrastive Learning ($S^2$GCL), a novel framework that integrates graph spectral information into spiking membrane dynamics, to bridge the gap between SNNs and graph-based contrastive learning.

- We develop a Spectrum-aware Membrane Potential (SaMP) mechanism by enriching membrane potentials based on eigenvalue encoding. Based on SaMP, we then derive a new SNN to encourage global structure-aware and more expressive spike learning.

- We introduce Overlapped Channel Grouping (OCG) to construct spike trains from static data with enhanced cross-channel dependencies. A dual-level contrast is used to enforce cross-view node-wise alignment and channel-wise alignment via group correlations.

Experiments on several benchmark datasets demonstrate the effectiveness, efficiency and high expressive capacity of our proposed $S^2$GCL.

## 2 RELATED WORKS

### 2.1 TRADITIONAL GRAPH CONTRASTIVE LEARNING

Graph contrastive learning (GCL) has emerged as a powerful paradigm for self-supervised representation learning on graph-structured data (Ju et al., 2024). Existing methods are generally designed around two key dimensions, augmentation strategies and contrastive objectives. For data augmentation, GRACE (Zhu et al., 2020) presents a hybrid scheme to augment graph views from both node attribute and structure levels. GCA (Zhu et al., 2021) develops adaptive edge perturbation based on centrality measures for contrastive learning. EPAGCL (Xu et al., 2025) reveals the superiority of dropping edges over adding edges for preserving structural semantics. For contrastive objective, CCA-SSG (Zhang et al., 2021) employs canonical correlation analysis to align embeddings instead of maximizing mutual information. SUGRL (Mo et al., 2022) designs two triple losses to increase inter-class variation and reduce intra-class variation. GGD (Zheng et al., 2022) reformulates contrastive learning as group discrimination to enhance computational efficiency.

## 2.2 Spiking Neural Networks on Graph

SNNs have gained significant attention in graph learning as an energy-efficient alternative. For example, SpikingGCN (Zhu et al., 2022) first integrates spiking neurons with graph convolution by treating SNNs as a simple binary module. SpikeNet (Li et al., 2023) adopts SNNs to efficiently capture the evolving dynamics of temporal graph data. COS-GNN (Yin et al., 2024) incorporates SNNs with Continuous Graph Neural Networks (CGNNs) to enhance information preservation and reduce information loss during SNNs propagation. DRSGNN (Zhao et al., 2024) introduces dynamic reactive spiking for temporal graphs by leveraging random-walk positional encoding as additional node features. MSG (Sun et al., 2024) explores spiking GNNs on Riemannian manifolds and designs a new training algorithm based on manifold differentiation. SPIKEGCL (Li et al., 2024) adopts the integrated encoder of GNN and SNN to learn binarized representations for contrastive learning.

## 3 Preliminaries

**Graph Neural Networks (GNNs).** Let $\mathcal{G}(\mathbf{A}, \mathbf{X})$ be an input graph, where $\mathbf{X} = [x_1, x_2 \ldots x_n] \in \mathbb{R}^{n \times d}$ denotes the node features and $\mathbf{A} \in \{0, 1\}^{n \times n}$ represents the adjacency matrix. The goal of GNNs is to learn node or graph-level representations by aggregating the information from its neighbors. Formally, the message aggregation at the $l$-th layer can be generally formulated as

$$h_i^{(l)} = \text{AGGREGATE}^{(l)} \left( h_i^{(l-1)}, h_j^{(l-1)}, \mathbf{A}_{ij} \right), \; j \in \mathcal{N}(i) \tag{1}$$

where $h_i^{(l)}$ is the hidden representation with $h_i^{(0)} = x_i$. $\mathcal{N}(i)$ represents the neighborhood of node $i$. By stacking multiple layers, GNNs can capture multi-hop structural patterns for node classification, link prediction and graph-level representation learning.

**Graph Contrastive Learning (GCL).** Given two contrastive views $\mathcal{G}$ and $\tilde{\mathcal{G}}$ generated by augmentation function $\mathcal{T}$, GCL aims to learn graph encoder $f_\theta$ by maximizing the following objective,

$$\max_\theta \; I(f_\theta(\mathcal{G}); f_\theta(\tilde{\mathcal{G}})) \tag{2}$$

where $I(\cdot)$ denotes the mutual information. For instance, let $z_i$ and $\tilde{z}_i$ be the node representation, we can compute the contrast loss between positive and negative pairs as follows (Xie et al., 2022),

$$\mathcal{L}_{con}(z_i, \tilde{z}_i) = -\log \frac{e^{\phi(z_i, \tilde{z}_i)/\tau}}{e^{\phi(z_i, \tilde{z}_i)/\tau} + \sum_{j \neq i} e^{\phi(z_i, z_j)/\tau} + e^{\phi(z_i, \tilde{z}_j)/\tau}} \tag{3}$$

where $\phi$ denotes the metric function and $\tau$ is a temperature constant.

**Spiking Neuron Models.** SNNs transmit information based on sparse spike trains, which generally contain integrate, fire and reset operations (Fang et al., 2021). To be specific, spike neurons first integrate current $c^{(t)}$ in the capacitor to accumulate potential, which can be formally written as

$$u^{(t)} = I(v^{(t-1)}, c^{(t)}), \quad u^{(t)} \in \mathbb{R}^{n \times d}, \; v^{(0)} \in \{0\}^{n \times d} \tag{4}$$

where $v^{(t-1)}$ is the membrane potential of time step $t-1$ and $u^{(t)}$ is the membrane potential after integration at the time step $t$. For example, the Integrate-and-Fire (IF) model simply defines the integration process as $u^{(t)} = v^{(t-1)} + c^{(t)}$. Then, fire operation can be triggered when $u^{(t)}$ exceeds a preset threshold $v_{th}$,

$$s^{(t)} = F(u^{(t)} - v_{th}), \quad s^{(t)} \in \{0, 1\}^{n \times d} \tag{5}$$

where $F(\cdot)$ is usually defined as the Heaviside function to fire a spike, i.e., $F(x) = 1$ if $x \geq 0$ and 0 otherwise. Finally, we need to reset the membrane potential, which can be expressed as

$$v^{(t)} = R(u^{(t)}, s^{(t)}, v_{th}), \quad v^{(t)} \in \mathbb{R}^{n \times d} \tag{6}$$

There are two types of reset, i.e., hard-reset (Fang et al., 2021) and soft-reset (Zhu et al., 2022). In this paper, we adopt the hard-reset defined as $u^{(t+1)} = u^{(t)}(1 - s^{(t)})$ with $v_{reset} = 0$.

## 4 Methodology

In this section, we will introduce a novel spectrum-enhanced spiking graph contrastive learning ($S^2$GCL) method, as shown in Figure 2. Similar to regular GCL architecture (Xie et al., 2022), our $S^2$GCL contains three components, including data augmentation, spectrum-aware SGNN encoder and contrastive objective.

Figure 2: The architecture of S$^2$GCL framework, which contains three components: (1) *Data augmentation* that generates different views; (2) *Spectrum-aware SGNN encoder* with Overlapped Channel Grouping (OCG) and Spectrum-aware Membrane Potential (SaMP) mechanisms; (3) *Dual-level contrastive objective* combining node-wise and channel-wise alignment.

## 4.1 DATA AUGMENTATION

Given an input graph $\mathcal{G} = (\mathbf{A}, \mathbf{X})$, we then generate the contrastive view $\tilde{\mathcal{G}} = (\tilde{\mathbf{A}}, \tilde{\mathbf{X}})$ by augmented function $\mathcal{T}(\cdot)$. For structural augmentation, we randomly remove edges to perturb graph connectivity while preserving global topology. Specifically, for adjacency matrix $\mathbf{A}$, we sample a masking matrix $\mathbf{M}_A \in \{0, 1\}^{n \times n}$ where each entry $m_{ij} \sim \text{Bernoulli}(p)$ and $p$ represents the drop probability. The function $\mathcal{T}(\mathbf{A}, \mathbf{X})$ on structural augmentation is defined as

$$\tilde{\mathbf{A}} = \mathcal{T}_{struc}(\mathbf{A}, \mathbf{X}) = \mathbf{A} \odot \mathbf{M}_A \tag{7}$$

where $\odot$ denotes Hadamard product. For feature augmentation, we randomly permute node features along the channel dimension to change local feature semantics. The function $\mathcal{T}(\mathbf{A}, \mathbf{X})$ on feature augmentation is formulated as

$$\tilde{\mathbf{X}} = \mathcal{T}_{feat}(\mathbf{A}, \mathbf{X}) = \mathbf{X}[:, \mathbf{P}] \tag{8}$$

where $\mathbf{P}$ is the random permutation indicator to shuffle channels. Using the above augmentations in Eqs.(7,8), we obtain two contrastive views, denoted as $\mathcal{G} = (\mathbf{A}, \mathbf{X})$ and $\tilde{\mathcal{G}} = (\tilde{\mathbf{A}}, \tilde{\mathbf{X}})$. Next, we adopt the proposed spectrum-aware SGNN as graph encoder to conduct representation learning.

## 4.2 SPECTRUM-INDUCED SGNN ENCODER

To obtain binary representations, we first construct spiking sequences via overlapped channel grouping. Based on sequential inputs, we then transform continuous features into spike trains via a novel SGNN encoder by integrating spectral information into membrane potential learning.

### 4.2.1 OVERLAPPED CHANNEL GROUPING

SNNs typically process sequential data based on a time-step mechanism. Consequently, a key question arises: how to construct sequential inputs for graph data to meet the sequential nature of SNNs. One commonly used approach is to directly repeat graph features multiple times, as employed in previous works (Zhu et al., 2022; Zhao et al., 2024). Another solution in SPIKEGCL (Li et al., 2024) is to partition node features into different groups. However, the former way inevitably leads to high computational and memory costs, while the latter fails to preserve cross-channel dependencies and weakens feature interactions.

We design an Overlapped Channel Grouping (OCG) strategy to construct spike trains across channel space. OCG explicitly preserves feature continuity across time steps, capturing temporal dependencies while enhancing intrinsic feature interactions among spike sequences. Specifically, we apply a sliding window approach to achieve overlapping division. Let $T$ be the time steps or the number of windows, features $X \in \mathbb{R}^{n \times d}$ can be divided into the following $T$ groups,

$$\mathbf{X} = [\mathbf{X}^{(1)}, \dots, \mathbf{X}^{(T)}], \quad \mathbf{X}^{(t)} \in \mathbb{R}^{n \times w}, t = 1 \dots T \tag{9}$$

where $w$ denotes the window size and $w << d$. $T$ is computed as $T = \lfloor \frac{d-w}{\alpha} \rfloor + 1$ with the sliding stride $\alpha$. Then, for views $\mathcal{G}$ and $\tilde{\mathcal{G}}$, we can have sequential inputs based on $T$ subgraphs as follows,

$$\mathcal{G} = \left[ \mathcal{G}^{(t)}(\mathbf{A}, \mathbf{X}^{(t)}) \right]_{t=1}^{T}, \quad \tilde{\mathcal{G}} = \left[ \mathcal{G}^{(t)}(\tilde{\mathbf{A}}, \tilde{\mathbf{X}}^{(t)}) \right]_{t=1}^{T} \tag{10}$$

Note that, the overlap between consecutive groups (i.e., $\mathcal{G}^{(t)} \cap \mathcal{G}^{(t+1)} \neq \emptyset$) establishes temporal dependencies in spike trains, enabling the input graph data to benefit from SNN's sequential processing.

### 4.2.2 SPECTRUM-GUIDED SNN

SNNs typically set the initial membrane potential (IMP) $v^{(0)}$ to zero (in Eq.(4)), limiting spike diversity (Shen et al., 2024). Figure 1(b) shows the spike firing patterns generated by IF (Abbott, 1999) model under constant input intensity. Here, the new patterns emerge by adjusting IMP from $0$ to $0.25$, which shows that the number of spiking patterns varies and thus pattern diversity increases depending on the IMP. Moreover, it is crucial for graph representation learning to take advantage of graph information (such as spectral or positional information). However, existing SNN-based GNNs usually regard SNNs as binary encoders, ignoring the exploration of graph information in spiking neuron dynamic learning. Therefore, an intuitive question naturally arises: *can we develop a novel SNN by leveraging spectrum to adaptively adjust IMP and thus induce SNN membrane dynamics?*

In this paper, we consider to integrate the spectrum of graph into SNN learning, which encourages the model to capture global structural information of graph. To be specific, let $\mathbf{L}$ be the Laplacian matrix of graph $\mathcal{G}$ with $\mathbf{L} = \mathbf{I} - \mathbf{D}^{-\frac{1}{2}} \mathbf{A} \mathbf{D}^{-\frac{1}{2}}$, we obtain the eigenvalues of $\mathbf{L}$, denoted as $\lambda_1, \lambda_2, \ldots, \lambda_n$ for $n$ nodes. We then use positional encoding $\rho(\lambda_i)$ to map eigenvalues to high-dimensional vectors as follows (Bo et al., 2023),

$$\rho(\lambda_i, 2j) = \sin\left(\epsilon\lambda_i/10000^{2j/f}\right); \quad \rho(\lambda_i, 2j+1) = \cos\left(\epsilon\lambda_i/10000^{2j/f}\right) \tag{11}$$

where $j \in \{0, 1, \ldots, f/2 - 1\}$. $\epsilon$ scales the frequency and $f$ is the channel dimension. Then, the learnable initial membrane potential $\hat{v}_i^{(0)}$ for node $i$ can be obtained as,

$$\hat{v}_i^{(0)} = \phi_{\text{init}}\left(\lambda_i || \rho(\lambda_i)\right) \tag{12}$$

where $\phi_{\text{init}}$ is a linear projection. Finally, Eq.(5) can be reformulated based on the learnable IMP as

$$u^{(t)} = I(\hat{v}^{(t-1)}, c^{(t)}), \quad u^{(t)} \in \mathbb{R}^{n \times f}, \quad \hat{v}^{(0)} \in \mathbb{R}^{n \times f} \tag{13}$$

where $\hat{v}^{(0)}$ continuously injects spectral biases during integration and extends from zero to the real number space $\mathbb{R}$. In this paper, we call the definition Eq.(13) as Spectrum-aware Membrane Potential (SaMP). Based on SaMP, we can derive our Spectrum-guided Spiking GNN as follows,

$$\text{Learnable IMP: } \hat{v}_i^{(0)} = \phi_{\text{init}}(\lambda_i \| \rho(\lambda_i)), \quad \phi_{\text{init}} \in \mathbb{R}^{(f+1) \to f}$$

$$I(t): \quad u_i^{(t)} = \hat{v}_i^{(t-1)} + \frac{1}{\tau}\left(h_i^{(t,L)} - (\hat{v}_i^{(t-1)} - v_{reset})\right)$$

$$F(t): \quad s_i^{(t)} = \begin{cases} 1, & u_i^{(t)} \geq v_{th}, \\ 0, & \text{otherwise.} \end{cases} \tag{14}$$

$$R(t): \quad \hat{v}_i^{(t)} = (1 - s_i^{(t)}) \cdot u_i^{(t)} + v_{reset} \cdot s_i^{(t)}$$

$$O(t): \quad v_{th} = \beta v_{th} + \gamma s_i^{(t)}$$

where $\tau$ is time constant. $O(t)$ is the threshold update function (Li et al., 2023) and $s_i^{(t)}$ is the spike output at time step $t$.

**Remark.** Here, we can also exploit some other graph information for dynamic membrane learning, such as positional or structural information (Rampášek et al., 2022), as shown in Table 2.

### 4.2.3 BINARY GRAPH REPRESENTATION LEARNING

Based on the above spiking sequences of two views, we further adopt the proposed spectrum-aware SGNN as graph encoder for binary representation learning. Due to the same learning process of each view, we temporarily omit the view $\tilde{\mathcal{G}}$ and provide the representation learning process defined on $\mathcal{G}$. For $\left[ \mathcal{G}^{(1)} \cdots \mathcal{G}^{(T)} \right]$, we construct a set of $T$ GNNs $f_\Theta(\cdot) = [f_{\Theta^{(1)}}^{(t)}]_{t=1}^T$ with $\Theta = \{\Theta^{(1)} \cdots \Theta^{(T)}\}$. Here, we adopt a $L$-layer Graph Convolutional Network (Kipf & Welling, 2017) to conduct neighborhood aggregation. Thus, the message propagation in Eq.(1) is formulated as follows,

$$h_i^{(t,l)} = \text{ReLU}\Big( \sum_{j \in \mathcal{N}(i)} \Theta^{(t)} \tilde{\mathbf{A}}_{ij} h_j^{(t,l-1)} \Big) \tag{15}$$

where $h_i^{(t,0)} = x_i^{(t)}$ and $\tilde{\mathbf{A}} = \mathbf{D}^{-\frac{1}{2}} \mathbf{A} \mathbf{D}^{-\frac{1}{2}}$. $\Theta^{(t)} \in \mathbb{R}^{w \times f}$ is a layer-inter shared network parameter, $w << d$. Then, we adopt spectrum-aware SNN to achieve binary representation learning. Specifically, the output $\mathbf{H}^{(t,L)} = [h_1^{(t,L)} \cdots h_n^{(t,L)}] \in \mathbb{R}^{n \times f}$ serves as input to the spectrum-induced SNNs module at time-step $t$. Then, we transform continuous signals $h_i^{(t,L)}$ into binary spikes $s_i^{(t)}$ by using the above proposed SNN model in Eq.(14). By collecting historical outputs across all time steps, we obtain binary spike trains of all groups/steps $\{s^{(1)} \cdots s^{(T)}\}$. Finally, we merge all groups together to obtain final representation $\mathbf{S} = [s^{(1)} \| \cdots \| s^{(T)}]$, where $\|$ denotes concatenation operation.

### 4.3 DUAL CONTRASTIVE OBJECTIVE WITH PROJECTION

In this section, we introduce a dual-level contrastive loss to align node representations and channel groups across views, including node-wise contrast and channel-wise contrast. Note that, we adopt the parametric estimator by incorporating a shared predictor head to compute the contrastive objective (Xie et al., 2022). The predictor head based on a fully connected layer (FC) maps embeddings to latent space for contrastive loss computation, e.g., $z^{(t)} = \text{FC}(s^{(t)})$ for the view $\mathcal{G}$.

### 4.3.1 NODE-WISE CONTRAST

For node-wise contrast, we compute intra-group contrastive loss by using margin ranking. Let $\{z^{(1)} \cdots z^{(T)}\}$ and $\{\tilde{z}^{(1)} \cdots \tilde{z}^{(T)}\}$ be the final output sets of $\mathcal{G}$ and $\tilde{\mathcal{G}}$, the node-wise contrastive learning can be achieved as ,

$$\mathcal{L}_{\text{node}} = \frac{1}{n} \sum_{t=1}^T \sum_{i=1}^n \max\left( 0, \delta\left( z_i^{(t)}, \tilde{z}_i^{(t)} \right) - \delta\left( z_i^{(t)}, \tilde{z}_j^{(t)} \right) + m \right) \tag{16}$$

where $\delta(\cdot)$ is the metric function, and $m$ is the margin constant. This contrast pulls positive pairs $(z_i^{(t)}, \tilde{z}_i^{(t)})$ closer while pushing negatives $(z_i^{(t)}, \tilde{z}_j^{(t)})$ apart within $t$-th group.

### 4.3.2 CHANNEL-WISE CONTRAST

Since feature augmentation re-permutes node feature by channel shuffling, we conduct intra-group and inter-group contrast by concatenating node embeddings of $T$ groups. Then, let $\mathbf{Z} = [z^{(1)} \| \cdots \| z^{(T)}]$ and $\tilde{\mathbf{Z}} = [\tilde{z}^{(1)} \| \cdots \| \tilde{z}^{(T)}]$ be the collection of final outputs for $\mathcal{G}$ and $\tilde{\mathcal{G}}$, the channel-wise contrast (Zhuo & Tan, 2022) can be defined as follows,

$$\mathcal{L}_{\text{channel}} = \frac{1}{k} \sum_i^k -\left( \log \frac{\exp\big(\phi\big(\mathbf{Z}_{\cdot i}, \tilde{\mathbf{Z}}_{\cdot i}\big)/\tau\big)}{\sum\limits_{j=1, j \neq i}^k \exp\big(\phi\big(\mathbf{Z}_{\cdot i}, \tilde{\mathbf{Z}}_{\cdot j}\big)/\tau\big)} + \log \frac{\exp\big(\phi\big(\tilde{\mathbf{Z}}_{\cdot i}, \mathbf{Z}_{\cdot i}\big)/\tau\big)}{\sum\limits_{j=1, j \neq i}^k \exp\big(\phi\big(\mathbf{Z}_{\cdot j}, \tilde{\mathbf{Z}}_{\cdot i}\big)/\tau\big)} \right) \tag{17}$$

where $k$ is the channel dimension of $\mathbf{Z}$ and $\tilde{\mathbf{Z}}$. The subscript $\cdot i$ represents the i-th channel of all nodes. $\phi(\cdot)$ denotes the metric function such as cosine similarity and $\tau$ is temperature parameter. This aligns corresponding channel groups across views while distinguishing mismatched groups.

Finally, the total unsupervised training objective that combines both node-wise and channel-wise contrast can be written as follows,

$$\mathcal{L} = \mathcal{L}_{\text{node}} + \eta \cdot \mathcal{L}_{\text{channel}} \tag{18}$$

where $\eta$ is a hyper-parameter to balance two terms.

### 4.4 COMPARISON WITH RELATED WORKS

We briefly discuss the differences between $S^2$GCL and related works that focus on the integration of SNNs and GNNs, including SpikingGCN (Zhu et al., 2022), DRSGNN (Zhao et al., 2024), MSG (Sun et al., 2024) and SPIKEGCL (Li et al., 2024). First, SpikingGCN and SPIKEGCL simply adopt SNNs as binary models, which fail to explore the characteristics of graph data to improve SNN dynamics. Differently, $S^2$GCL exploits spectral information to meet the spiking dynamics from the perspective of membrane potential. Second, MSG (Sun et al., 2024) introduces random-walk positional encoding as auxiliary node features to enhance temporal awareness. However, this spiking design relies on external positional features rather than being explicitly tied to graph structure. Beyond directly regarding it as auxiliary features, $S^2$GCL embeds spectral information into the initial membrane potential, allowing SNNs to naturally encode global structural information. DRSGNN (Zhao et al., 2024) explores spiking GNNs on Riemannian manifolds by transmitting manifold projections into SNNs. In contrast, $S^2$GCL naturally injects eigenvalue-based spectral biases into IMP learning and thus encourages global structure-aware spiking dynamics without any manifold projection.

## 5 EXPERIMENTS

### 5.1 EXPERIMENTAL SETUPS

**Dataset Settings.** We conduct experiments on six benchmark datasets, including three citation graphs (Cora, CiteSeer, PubMed) (Sen et al., 2008), two co-purchase graphs (Photo, Computers) (Shchur et al., 2018) and one co-author graph (CS) (Shchur et al., 2018). For all datasets, we randomly partition the nodes into 80% for training, 10% for validation, and the rest 10% for test to construct data splits. For each dataset, we report the average results across five splits.

**Parameter Settings.** We adopt a two-layer model for binary graph representation learning, where GCN (Kipf & Welling, 2017) is used for message aggregation and PLIF model (Fang et al., 2021) is used for spike learning. We vary the window size $w$ within the range $[22, 122]$ and the stride $\alpha$ within the range $[10, 100]$ to fit the differences in channel number for different datasets. The initial fire threshold is tuned within $[3e - 4, 1]$ and the decay parameters $\beta$ and $\gamma$ are set to 0.2 and 0.7. In addition, the temperature $\tau$ is set to 0.1. The coefficient $\eta$ in Eq.(18) is tuned within $[0.2, 0.5]$ and the learning rate of training model is set to $1e - 3$ for all datasets. All experiments are conducted on an NVIDIA RTX 3090 GPU with 24 GB memory.

### 5.2 COMPARISONS

**Baselines.** We compare $S^2$GCL against some state-of-the-art methods, which can be divided into the following five types: full-precision GNNs: GCN (Kipf & Welling, 2017) and GAT (Veličković et al., 2018); spiking-based GNNs: SpikingGCN (Zhu et al., 2022), DRSGNN (Zhao et al., 2024) and MSG (Sun et al., 2024); 1-bit quantized GNNs: BANE (Yang et al., 2018), Bi-GCN (Wang et al., 2021) and BinaryGNN (Bahri et al., 2021); full-precision GCL methods: GRACE (Zhu et al., 2020), CCA-SSG (Zhang et al., 2021), SUGRL (Mo et al., 2022), GGD (Zheng et al., 2022) and EPAGCL (Xu et al., 2025); one spiking GCL method SPIKEGCL (Li et al., 2024).

**Main Results.** Table 1 reports the comparison results of all methods. Overall, $S^2$GCL achieves the best performance on all datasets. Specifically, we have the following observations: (1) $S^2$GCL outperforms traditional GNNs, which shows that spectrum-aware spiking dynamics enables more effective graph representation learning. (2) $S^2$GCL obtains higher performance than binarized GNNs, which shows the advantage of spiking-driven mechanism on binary representation learning. (3) Compared to pure SNN-based models (SpikingGCN and SPIKEGCL), $S^2$GCL shows consistent improvement. This shows that it is helpful for GCL by explicitly injecting graph spectrual information rather than treating SNNs as generic binary encoders. (4) Compared with DSRGNN and MSG, which use graph information as auxiliary feature-level input , $S^2$GCL achieves better results on all datasets. This shows that integrating graph information into dynamic membrane learning is superior to merely conducting feature-level enhancement or concatenation on the original features.

**Other Results.** In Eqs.(12-13), we propose SaMP by introducing the spectrum of eigenvalue into IMP learning. While graph data also contain other rich information, such as local or global posi-

Table 1: Comparison results of node classification. **U**: Unsupervised GNNs; **S**: Spiking GNNs; **B**: Binary GNNs.

| Method | U | S | B | Cora | CiteSeer | PubMed | Photo | Computers | CS |
|---|---|---|---|---|---|---|---|---|---|
| GCN | | | | 86.81±1.06 | 73.49±0.45 | 83.08±1.09 | 93.17±0.38 | 83.40±0.47 | 88.39±0.42 |
| GAT | | | | 83.48±1.33 | 76.14±1.93 | 84.90±0.59 | 90.38±0.81 | 82.57±1.13 | 91.75±0.52 |
| BANE | | | ✓ | 79.19±1.61 | 68.01±2.74 | 82.40±1.13 | 82.20±0.53 | 77.37±1.62 | 88.50±0.65 |
| Bi-GCN | | | ✓ | 86.40±1.21 | 75.48±2.08 | 86.61±0.52 | 94.52±0.79 | 90.38±0.31 | 90.12±0.78 |
| BinaryGNN | | | ✓ | 84.52±1.57 | 73.47±1.94 | 86.65±0.47 | 94.09±0.66 | 90.09±0.43 | 93.52±0.22 |
| SpikingGCN | | ✓ | | 86.44±1.27 | 75.66±0.01 | 85.15±0.49 | 90.60±0.82 | 83.24±0.87 | 92.05±0.58 |
| DRSGNN | | ✓ | | 86.96±1.39 | 72.95±1.84 | 76.06±0.01 | 92.15±0.64 | 85.90±0.38 | 92.82±0.33 |
| MSG | | ✓ | | 87.04±1.39 | 75.60±1.26 | 85.41±0.38 | 93.75±0.73 | 86.85±0.26 | 94.09±0.73 |
| GRACE | ✓ | | | 86.89±0.80 | 76.57±1.71 | 86.25±0.49 | 93.99±0.67 | 89.28±0.49 | 91.56±0.85 |
| CCA-SSG | ✓ | | | 86.44±1.49 | 75.06±1.99 | 84.81±0.92 | 93.41±0.81 | 89.70±0.67 | 93.16±0.32 |
| SUGRL | ✓ | | | 86.59±1.20 | 73.91±1.70 | 85.53±0.76 | 93.90±0.66 | 89.12±0.52 | 93.99±0.57 |
| GGD | ✓ | | | 85.18±1.60 | 74.66±1.53 | 85.37±0.62 | 93.02±0.25 | 88.68±0.26 | 92.76±0.11 |
| EPAGCL | ✓ | | | 84.30±0.47 | 71.24±0.45 | 83.85±0.25 | 91.88±0.20 | 87.95±0.26 | 89.17±0.34 |
| SPIKEGCL | ✓ | ✓ | | 87.36±1.00 | 75.47±2.60 | 87.61±0.32 | 94.01±0.62 | 90.90±0.28 | 93.80±0.57 |
| OURS | ✓ | ✓ | | **88.12±0.57** | **76.63±2.30** | **87.63±0.60** | **94.79±0.80** | **91.29±0.25** | **94.24±0.70** |

tional/structural information. Therefore, we can naturally extend IMP by injecting different forms of graph information to guide SNN membrane dynamics. Table 2 reports the results of SaMP with three types of graph information (raw and encoding versions): eigenvalues (Eig), Laplacian feature (Lap) and random walk (RW). The results show that our method is flexible and thus can incorporate diverse graph information to enhance SNNs' learning.

Table 2: Results of different graph information (Graph Info.).

| Graph Info. | Raw Lap | Raw RW | Lap encoding | RW encoding | Eig encoding |
|---|---|---|---|---|---|
| Cora | 87.83 ± 1.15 | 87.59 ± 0.96 | 87.99 ± 1.38 | 88.07 ± 0.61 | 88.12 ± 0.57 |
| Photo | 94.24 ± 0.75 | 94.10 ± 0.54 | 94.36 ± 0.47 | 94.58 ± 0.91 | 94.79 ± 0.80 |
| Citeseer | 76.33 ± 1.85 | 76.38 ± 2.08 | 76.04 ± 2.05 | 75.90 ± 2.60 | 76.63 ± 2.30 |
| Computers | 91.25 ± 0.26 | 91.11 ± 0.35 | 91.14 ± 0.50 | 91.13 ± 0.49 | 91.29 ± 0.25 |

**Ablation Study.** We conduct ablation study to evaluate the effectiveness of core components in $S^2$GCL. There are four variants: $S^2$GCL w/o SaMP, $S^2$GCL w/o OCG, $S^2$GCL w/o Channel contrast and $S^2$GCL w/o All. From Figure 3, one can find that: (1) Removing the SaMP mechanism leads to a clear performance degradation, demonstrating the benefits of spectral information in improving SNNs' expressive capacity. (2) Removing OCG and channel contrast also degrades model performance, which shows their effectiveness in enhancing discriminative feature learning.

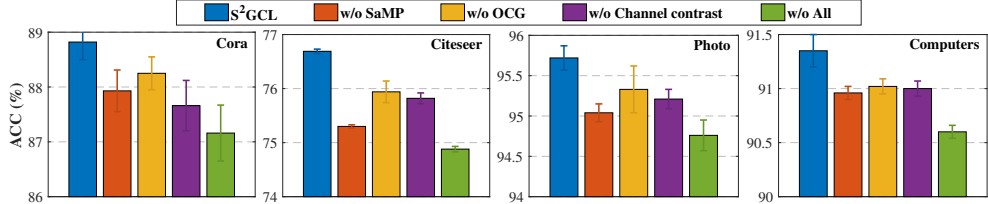

Figure 3: The ablation study of $S^2$GCL. w/o denotes removing a component from $S^2$GCL.

**Parameter Analysis.** We first analyze the time step $T$ and overlapping scale $\pi$, as shown in Figure 4. Figure 4(a)-(b) show the effect of time step $T$. One can note that as $T$ increases, the classification accuracy gradually improves while the running time also increase. Therefore, we adjust $T$ to achieve a balance between performance and efficiency. Figure 4(c) reports the classification results with

different ratio $\pi$ (defined as $\frac{w-\alpha}{w}$). Lower ratios weaken temporal correlations while higher ratios increase computation. Therefore, we take an appropriate ratio for balance.

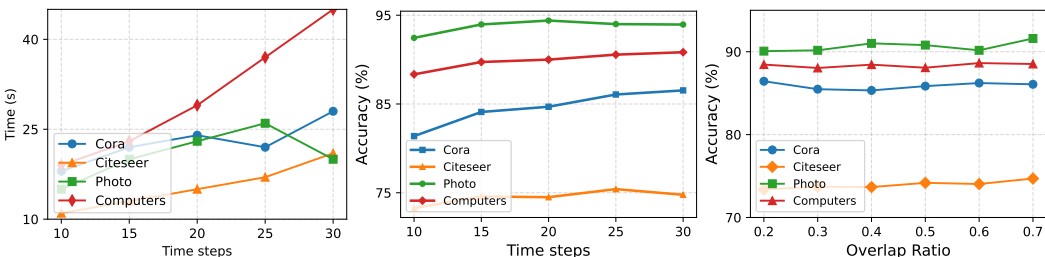

(a) Running time with different $T$    (b) Test accuracies with different $T$    (c) Test accuracies with different $\pi$

Figure 4: Parameter analysis of time step $T$ and overlap ratio $\pi$.

**Efficiency Analysis.** In this section, we follow SPIKEGCL (Li et al., 2024) and report the comparisons of energy consumption on Photo and Computers datasets, as shown in Figure 5. One can observe that SNN-based methods generally require less energy consumption than ANN-based methods. $S^2GCL$ achieves the lower energy cost than most SNN-based models. Besides, compared with SPIKEGCL, $S^2GCL$ can obtain competitive energy cost with better performance.

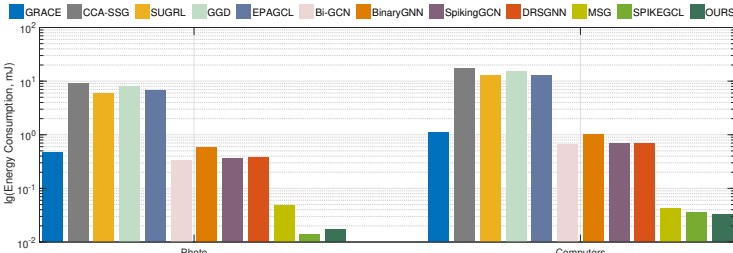

Figure 5: Comparison results of energy consumption on Photo and Computers datasets.

**Intuitive Demonstration.** Here, we follow the previous work (Shen et al., 2024) and provide a toy example to show the advantage of spectrum-aware IMP. Given a node $x_i \in \mathbb{R}^4$, we divide $x_i$ into 4 groups along channel space to construct the spike inputs of 4 time steps. As shown in Figure 6, the horizontal and vertical axes respectively correspond to all possible spike patterns accepted and fired by IF model (Abbott, 1999). The white square denotes a pattern mapping, meaning that the neuron accepts a spike pattern from the horizontal axis and fires correspond spike pattern on the vertical axis. We note that spectrum-aware IMP obtain more diverse pattern mappings than fixed IMP (usually set to zero), which demon-

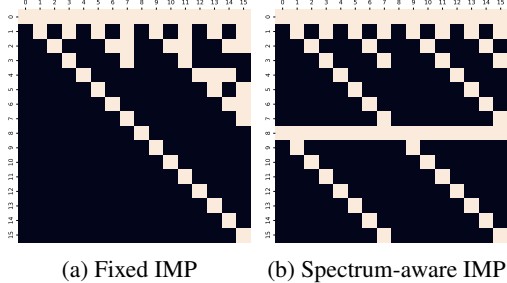

(a) Fixed IMP    (b) Spectrum-aware IMP

Figure 6: A toy visualization of pattern mapping by IF model with 4 time steps.

strates that our proposed spectrum-aware IMP can enhance the expressive power of spiking neurons.

## 6 CONCLUSION

This paper proposes a novel $S^2GCL$ method, which embeds graph spectral information into spiking membrane dynamics. The core of $S^2GCL$ is to develop a new spectrum-aware SNN by using eigenvalue-based membrane learning to improve the expressive capacity of spiking neurons. Also, we introduce overlapped channel grouping to construct spike trains for graph data and adopt channel-wise contrast to enhance cross-view representation learning. Extensive experiments validate the effectiveness of $S^2GCL$. In the future, we will further explore more perspectives for the integration of spiking dynamics and graph structure information to develop new spiking graph neural networks.

ETHICS STATEMENT

This research does not involve human subjects, personally identifiable information, or sensitive data. All datasets used in our experiments are publicly available and widely used in graph-based representation learning field. The proposed method is intended for academic research purposes and does not pose foreseeable risks of misuse. We confirm that our study complies with the ICLR Code of Ethics.

REPRODUCIBILITY STATEMENT

Detailed descriptions of dataset settings, model architecture and parameter settings are provided in Section 5.1. More detail information is listed in Appendix.

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

## APPENDIX

## A  OVERALL ALGORITHM FLOW

Here, we provide the pseudocode of our S$^2$GCL framework in Algorithm 1 to facilitate understanding.

---

**Algorithm 1** Learning process of S$^2$GCL

---

**Require:** Graph $\mathcal{G} = (\mathbf{A}, \mathbf{X})$, the window size $\omega$, the sliding stride $\alpha$, the time steps $T$; the encoder $f_\Theta(\cdot) = [f_{\Theta^{(1)}}^{(t)}]_{t=1}^T$ and the predictor head FC$(\cdot)$ with learnable parameter $\Omega$
**Ensure:** Optimized encoder $f_\Theta$

1: **while** not converged **do**
2:      $\tilde{\mathcal{G}}(\tilde{\mathbf{A}}, \tilde{\mathbf{X}}) = \mathcal{T}(\mathcal{G})$
3:      $\mathcal{G} = \left[\mathcal{G}^{(t)}(\mathbf{A}, \mathbf{X}^{(t)})\right]_{t=1}^T, \quad \tilde{\mathcal{G}} = \left[\mathcal{G}^{(t)}(\tilde{\mathbf{A}}, \tilde{\mathbf{X}}^{(t)})\right]_{t=1}^T$
4:      **for** $t = 1$ to $T$ **do**
5:          $\mathbf{s}^{(t)} = f_{\Theta^{(t)}}^{(t)}(\mathcal{G}^{(t)})$
6:          $\mathbf{z}^{(t)} = \text{FC}_\Omega(\mathbf{s}^{(t)})$
7:          $\tilde{\mathbf{s}}^{(t)} = f_{\Theta^{(t)}}^{(t)}(\tilde{\mathcal{G}}^{(t)})$
8:          $\tilde{\mathbf{z}}^{(t)} = \text{FC}_\Omega(\tilde{\mathbf{s}}^{(t)})$
9:          Calculate dual-wise contrastive loss $\mathcal{L}$ based on $\mathbf{z}^{(t)}, \tilde{\mathbf{z}}^{(t)}$
10:          Update $\Theta = \{\Theta^{(1)} \cdots \Theta^{(T)}\}, \Omega$
11:      **end for**
12: **end while**
13: **return** $f_\Theta(\cdot)$

---

## B  EXPERIMENTAL SETUP

**Datasets.** We conduct experiments on six benchmark datasets, including three citation graphs (Cora, CiteSeer, PubMed) (Sen et al., 2008), two co-purchase graphs (Photo, Computers) (Shchur et al., 2018) and one co-author graph (CS) (Shchur et al., 2018). The detailed information is listed in Table 3. We randomly partition the nodes into 80% for training, 10% for validation, and the rest 10% for test to generated five data splits for each dataset.

Table 3: Dataset statistics.

| Dateset | Cora | CiteSeer | PubMed | Photo | Computers | CS |
|---|---|---|---|---|---|---|
| Features | 1,433 | 3,703 | 500 | 745 | 767 | 6,805 |
| Nodes | 2,708 | 3,327 | 19,717 | 7,650 | 13,752 | 18,333 |
| Edges | 10,556 | 9,104 | 88,648 | 238,162 | 491,722 | 163,788 |
| Classes | 7 | 6 | 3 | 8 | 10 | 15 |

**Implement details.** We follow the previous work (Li et al., 2024) and implement our model with PyTorch platform and PyTorch Geometric library (Fey & Lenssen, 2019). All datasets used in this

paper are also publicly available in the PyTorch Geometric library. All experiments are conducted on an NVIDIA RTX 3090 Ti GPU with 24 GB memory unless specified. For comparison methods, all the results are obtained according to the experimental settings provided by the authors and then tuned in our experiments to achieve better performance.

## C  THE USE OF LARGE LANGUAGE MODELS (LLMS)

In this paper, Large Language Models (LLMs) are used exclusively as auxiliary aids for language refinement. In particular, we utilize LLMs to improve the phrasing and clarity of the manuscript without modifying any technical content or experimental findings. To achieve this, we employ a simple prompt, such as `"Please revise this statement for better clarity and fluency."`, to enhance readability. No LLM-generated content related to methodology design, implementation or experimental results is included in this paper.

