# OpenReview forum: "Spectrum-aware Spiking Dynamics for Graph Contrastive Learning"
_ICLR.cc/2026/Conference — ICLR 2026 Conference Withdrawn Submission_

### Official Review · Reviewer_S6tV · 2025-10-31

**Soundness:** 3
**Presentation:** 3
**Contribution:** 2
**Rating:** 4
**Confidence:** 4

**Summary:**

The study identifies two main limitations of existing spiking graph neural networks: the failure to capture intrinsic graph properties and the limited diversity of spike firing patterns. To address the above challenges, it proposes a Spectrum-inspired Spiking Graph Contrastive Learning ($S^2GCL$) framework which effectively introduces graph information into spiking neural dynamics. It consists of spectrum-aware membrane potential (SaMP) and overlapped channel grouping (OCG) modules. In addition, the framework constructs a dual-level contrastive loss to align both node-wise and channel-wise representations. Experiments on six benchmark datasets demonstrate that $S^2GCL$ achieves improved accuracy and promising energy efficiency compared to full-precision, binary and spiking GNN baselines.

**Strengths:**

1. The proposed SaMP mechanism is intuitive, which explores the diversity of spike patterns induced by initial membrane potential variations.
2. The experimental evaluation encompasses multiple datasets and a broad range of baselines including binary, spiking, and unsupervised GNNs. The experimental results show that $S^2GCL$ consistently outperforms prior methods in terms of accuracy.
3. The parameter analysis and energy consumption comparisons provide empirical evidence supporting the feasibility of this work for some resource-constrained applications.

**Weaknesses:**

1. $S^2GCL$ appears to offer an incremental improvement compared with existing studies. The grouped subgraphs in OCG are largely derived from the paradigm proposed in SpikeGCL. SaMP, which incorporates positional/structural information into membrane potential, is closely related to the GNNs with positional/structural embeddings.
2. In Equation 14, the design of Learnable IMP aims at adding positional/structural information to initial membrane potentials, $u_i=\frac{1}{\tau}h_i+(1-\frac{1}{\tau})\tilde{v}_{pe/se}$. It seems to just transfer the widely used PE/SE into the spiking neurons as the initial membrane potential, without introducing a fundamentally new mechanism for dynamic membrane updating. A clearer distinction from existing PE/SE-based GNNs would strengthen the novelty claim.
3. The paper lacks a detailed discussion of the computational cost associated with graph spectral information. Since eigenvalue computation can be expensive for large graphs, this operation may become a major bottleneck. The experiments are limited to small-to-medium graphs. It raises my concern whether the spectrum-aware spiking neurons hinder the scalability of $S^2GCL$ to large-scale graphs (e.g., OGB datasets).

**Questions:**

1. Could the authors provide more experimental results on large-scale graph datasets to verify the scalability of the proposed framework?
2. The spectrum-aware spiking neurons are built upon LIF rather than spiking variants with learnable threshold potentials. Given that the initial membrane potential is influenced by graph information, how sensitive is the proposed model to the choice of threshold membrane potentials?
3. In the efficiency analysis, is the computational overhead of pre-calculating graph spectral information considered? It would be helpful to include a more detailed complexity analysis or a formulated energy consumption estimation to better quantify the energy gaps among different baselines.

---

### Official Review · Reviewer_3Pop · 2025-11-01

**Soundness:** 2
**Presentation:** 1
**Contribution:** 1
**Rating:** 2
**Confidence:** 3

**Summary:**

The paper proposes S^2GCL, a spiking graph contrastive learning framework that aims to reduce computational energy while preserving representation quality. To enhance the expressive power of spike learning, the paper introduces Overlapped Channel Grouping (OCG) for sequential aspects and Spectrum-aware Membrane Potential (SaMP) for graph structural aspects. On six small to medium benchmarks (citation datasets and Amazon datasets), S^GCL reports accuracy improvements over full-precision GCL baselines, binary GNNs, and prior spiking GNN/GCL approaches, while also demonstrating energy reductions compared with ANN baselines.

**Strengths:**

A substantive assessment of the strengths of the paper, touching on each of the following dimensions: originality, quality, clarity, and significance. We encourage reviewers to be broad in their definitions of originality and significance. For example, originality may arise from a new definition or problem formulation, creative combinations of existing ideas, application to a new domain, or removing limitations from prior results.

[1] Conceptual link between spectrum and spiking dynamics.
Since prior SGNNs treat SNNs as binarizers or inject positional features at the input, directly biasing membrane states with graph structure is a novel idea that could broaden the design space for spiking GNNs. The paper establishes a clear conceptual link between the graph’s spectral perspective and spiking dynamics.

[2] Practical procedure for building spike sequences on static graphs.
The proposed Overlapped Channel Grouping (OCG) with a sliding-window mechanism provides a pragmatic alternative to feature repetition or non-overlapped grouping. Moreover, it naturally aligns with channel-wise contrastive learning, enabling the use of a channel contrastive loss. The paper empirically demonstrates the effectiveness of OCG across several benchmarks.

[3] Rich empirical results and ablations.
On six commonly used datasets, S²GCL achieves competitive performance not only against other spike-based models but also against strong ANN baselines. The ablation and parameter studies (in Figs. 3–4) are informative and well aligned with the stated motivations.

**Weaknesses:**

[W1] Limited scope of evaluations and unclear experimental settings.
All datasets are small to medium in size and cannot effectively validate the proposed method. Since the overall architecture follows SpikeGCL [1], the paper should at least include results on the OGBN-Arxiv and OGBN-MAG datasets. SaMP claims global structure awareness and robustness to node re-indexing, spectral sign flips, and graph perturbations; these should be explicitly demonstrated.
The main concern is that baseline experimental settings are not clearly specified, and the main configuration of S²GCL remains ambiguous. For instance, in MSG, there exist several settings such as the manifold choice (e.g., Lorentz) and the learning rate. Because the reported performance gap is nearly identical to the second-best results (Table 1), the experimental settings should be clearly described. The authors should explicitly explain how the baseline results were obtained, at least in the appendix.
In addition, GNN performance is highly sensitive to the learning rate. The authors should explore multiple learning rate configurations. Currently, the main results appear to be evaluated using only a single learning rate (1e-3) across all datasets, as stated in line 355. Such unclear experimental settings make it difficult to assess the effectiveness of the proposed method.

[W2] The main effectiveness of the method is not clear.
In line 254, the spectrum-guided spiking GNN appears to combine several elements from prior works. Specifically, line 264 references adaptive threshold methods, and in the experimental setup, line 256 indicates substitution with PLIF neurons, which already yield significant improvements compared to the original LIF or IF neurons. For these reasons, the learnable IMP component seems incremental, mainly adding more parameters to enhance positional encoding.
It is not clear whether the observed improvements stem primarily from the learnable IMP or from a combination of previously established techniques (e.g., adaptive threshold + PLIF).

[W3] Lack of efficiency and overhead analysis
The paper lacks a detailed analysis of computational efficiency and overhead. Compared to prior work, S^2GCL, which uses T sequential GNNs, appears to incur substantial memory overhead. Additionally, the description of efficiency calculation (line 447) is unclear. At least, the paper should explain how the theoretical energy was computed in the appendix. Including metrics such as MAC/SOP and FLOPs for energy computation would provide a clearer and more quantitative understanding of efficiency.

[1] Li, Jintang, et al. "A Graph is Worth 1-bit Spikes: When Graph Contrastive Learning Meets Spiking Neural Networks." The Twelfth International Conference on Learning Representations.

**Questions:**

1.	How was the energy consumption in Figure 5 calculated for the S^2GCL? Are there any additional computational overheads in S^2GCL, or does Figure 5 already account for these overheads?
2.	The experimental settings are not clearly described. All baseline configurations and experimental details used to obtain the results in Table 1 should be fully elaborated.
3.	Why do the authors bias using IMP rather than appending spectral positional encodings (as in SAN-style methods) to the input features before the spiking neuron? Please compare with an otherwise identical model that adds spectral PEs to X and keeps $v^(0)=0$.
4.	Could the authors discuss the difference between the learnable threshold and IMP methods? Since the effectiveness of firing in SNNs depends on the difference between the threshold and membrane potential, learnable threshold methods could clarify the issue illustrated in Fig. 1(b).
5.	How exactly is energy computed (e.g., based on MAC vs. AC counts or scaling constants)? Please provide detailed information on the energy estimation procedure.
6.	Is S^2GCL feasible for deployment on neuromorphic chips? A discussion of the modifications required for neuromorphic implementation would strengthen the paper.

---

### Official Review · Reviewer_kSp5 · 2025-11-01

**Soundness:** 3
**Presentation:** 3
**Contribution:** 2
**Rating:** 2
**Confidence:** 3

**Summary:**

This paper introduces spiking graph domain adaptation (SGDA) and proposes DeSGraDA, a framework for graph classification under distribution shift. The method has three modules: (1) degree‑conscious spiking representation, which adapts neuron firing thresholds per node degree to balance firing across high/low‑degree nodes; (2) temporal distribution alignment, which summarizes membrane‑potential sequences with self‑attention and aligns source/target via an adversarial discriminator; and (3) pseudo‑label distillation. Experiments on SEED/BCI EEG, PROTEINS/DD proteins, and COX2/BZR chemistry datasets report improvements over graph and spiking baselines.

**Strengths:**

- Thorough experimental verification. Performance was thoroughly experimentally verified on various datasets.
- Theoretical support. They provide the generalization bound for SGDA.

**Weaknesses:**

- Limited novelty. DeSGDA comprises three components: node degree-aware personalized spiking representation, adversarial feature distribution alignment, and pseudo-label distillation. However, the second (domain alignment) and the third (pseudo labeling) components are well-known techniques in domain adaptation. Thus, their contribution is limited.
- Concerns about effectiveness. The overall experimental improvements are marginal.
- Unclear motivation. Additional explanation is needed for why domain adaptation should be solved with spiking neural networks.
- Lack of experimental verification of theoretical analysis (theorems)
- Overhead analysis. There is no analysis of the overhead caused by the proposed approach.

**Questions:**

Please refer to Weaknesses section.

---

### Note · Authors · 2025-11-13

I have read and agree with the venue's withdrawal policy on behalf of myself and my co-authors.